

# A calculation method of unsaturated soil water content based on
# thermodynamic equilibrium
Danhui Su[1], Jianwei Zhou[1,2], Zuocong Yin[3], Haibo Feng[1], Xiaoming Zheng[1], Xu Han[1], Qingqiu Hou[1]
[1]School of Environmental Studies, China University of Geosciences, Wuhan, 430085, China
[2]Key Laboratory of Mine Ecological Effects and Systematic Restoration, Beijing, 100081, China
[3]Centralsouth Bureau of China Metallurgical Geology Bureau, Wuhan, 430080, China
*Correspondence to*: Jianwei Zhou (jw.zhou@cug.edu.cn)
**Abstract.** Accurate determination of soil water content is crucial for studying the local ecological water cycle, agriculture,
forests and grasslands management, slope stability, environmental processes, and ecological problems. Current measurement
methods of soil water content have challenges with the complicated operation, high costs, significant system errors, and soil
type-specific accuracy. In this study, we proposed an alternative method to measure soil water content using the ratios of mass
and density of water to vapor based on system science and thermodynamics. We derived the mass ratio and density ratio of
water to vapor as a function of temperature from published data and validated the accuracy of the proposed method using
observed temperature, relative humidity, and volumetric water content in two soil textures of medium and fine sand. We further
demonstrated that the mass ratio function is independent of the nature of the water-containing media using theoretical and
statistical analysis. The absolute error of soil water content between the calculated and measured ones is less than 1% and is
positively correlated with temperature. This study significantly improves the measurement accuracy of soil water content,
eliminates the effect of water-containing medium on soil water content measuring, and has excellent potential for application
in the field soil and even rock water content measurement.
## 1 Introduction
Soil is crucial to life on Earth (Weil and Brady, 2017b), and soil water is one of the most critical factors. Determination of soil
water content is fundamental in studying soil's physical, chemical, and ecological processes. Soil water is an essential
component of the ecological water cycle, as it regulates rainfall infiltration, surface runoff, and evaporation, as well as affecting
river flow and local climate (Anh et al., 2015; Daly and Porporato, 2005; Good et al., 2015; Norton et al., 2022). Soil water
content plays an essential role in plant growth. It influences the germination and emergence of plant seeds (Nielsen et al.,
1995), as well as the health and quality of plants (Ma and Fan, 2020; Satoh and Kakiuchi, 2021), and has a direct impact on
forest and agricultural productivity (Breda et al., 2006). Soil water is the primary factor influencing microbial community
structure and enzyme activities (Brockett et al., 2012; Ni et al., 2022). On the one hand, it is the basis for microbial survival.
On the other hand, soil water content determines the air content and gas diffusion in the soil, thus affecting microbial activity
(Hillel, 2008a). The mechanical properties of soil, such as consistency, plasticity, shear strength, etc., depend on soil water





content (Hillel, 2008a; Lekshmi et al., 2014). Therefore, it significantly impacts geotechnical engineering structure and slope
stability (Ng Charles and Pang, 2000; Rahimi et al., 2010; Xu and Yang, 2018). In general, accurate determination of soil
water content is important in geology, hydrology, ecology, agronomy, engineering, and other fields (Ma et al., 2016).
Researchers have worked extensively on determining soil water content and have accomplished many critical results.
Nevertheless, there are some limitations. Measurement methods of soil water content are often classified as direct and indirect
(Shukla, 2013; Topp et al., 2007). The direct method is the thermo-gravimetric method, particularly the oven drying method.
The soil sample is dried in an oven at a constant temperature of about 105°C until the weight of the dry soil becomes constant,
and the difference in soil weight before and after drying is the soil water content (Pansu and Gautheyrou, 2006; Topp and Ferré,
2002). The method is commonly used to determine soil water content and is considered a standard. However, the sampling
and transportation process may cause disturbance of the soil structure and loss of moisture. Moreover, part of the water is not
entirely dried (Wang et al., 2011), and some organic matter may oxidize and decompose at 105°C (Hillel, 1980; Lekshmi et
al., 2014). Therefore, the method may produce a significant systematic error.
Indirect methods measure the soil's physical or chemical property that depends on its water content. They include the radiation
methods (neutron probe, gamma-ray attenuation, nuclear magnetic resonance) (Klenke and Flint, 1991; Manalo et al., 2003;
Strati et al., 2018), dielectric methods (time-domain reflectometry, frequency domain reflectometry, amplitude domain
reflectometry, capacitive technique) (Moroizumi and Sasaki, 2008; Noborio, 2001; Sheets and Hendrickx, 1995; Whalley et
al., 1992; Xu et al., 2012), remote sensing methods (microwave remote sensing and ground-penetrating radar) (Huisman et al.,
2003; Jackson, 2002; Liu et al., 2019; Njoku and Entekhabi, 1996), and optical methods (fiber optic sensor technique and near-
infrared optical technique) (Alessi and Prunty, 1986; Lekshmi et al., 2014; Lim et al., 2020; Robinson et al., 2008). Radiation
methods measure soil water content by the property of radioactive substances concerning soil moisture. Although accurate and
rapid, they are costly to use and have risks of radiation exposure (Jarvis and Leeds-Harrison, 1987). The dielectric methods
measure soil water content utilizing the dielectric properties determined by soil moisture. They are quick and repeatable, but
they ignore the impact of temperature on the soil's dielectric properties and have a considerable measurement error (Noborio,
2001). Remote sensing methods use electromagnetic energy emitted or reflected from the land surface. They can obtain the
soil water content over a large area, while they have a shallow measurement depth and low measurement accuracy. In addition,
the results could be influenced by the nature of the soil and vegetation cover (Bogena et al., 2015). Optical methods capitalize
on changes in the characteristics of the incident and reflected light passing through the soil to determine the soil water content.
The methods allow distributed remote measurements and are resistant to interference, but they are expensive, and the soil type
and surface roughness significantly impact measurement accuracy (Robinson et al., 2008). In summary, current measurement
methods of soil water content have challenges with the complicated operation, high costs, significant system errors, and soil
type-specific accuracy.
To address the issues mentioned above, we propose a new determination and calculation method for soil water content based
on system science and thermodynamics, which minimizes the impact of soil type and improves measurement accuracy. This
study presents the relationship between soil water content and the mass ratio and density ratio of water to vapor at





thermodynamic equilibrium. Sandbox experiments construct the mass ratio of water to vapor as a function of temperature.
Furthermore, the mass ratio function is independent of the nature of the water-containing media, according to theoretical
analysis and statistical results. The absolute error of soil water content between the calculated and measured is less than 1%
and is positively correlated with temperature.

## 69  2 Theory and calculation

### 70  2.1 Theory

#### 71  2.1.1 Thermodynamic equilibrium

Unsaturated soils consist of mineral particles and pores which contain water and air, with vapor water being a trace but
important component of the air. The total internal energy, volume, and amount of substance of the system are constant,
assuming that the water and vapor are in an isolated unary two-phase system and the surface of the mineral particles is the
boundary. According to the second law of thermodynamics and the fundamental equations of thermodynamics for open
systems (Cui, 2009), when the system is in thermodynamic equilibrium, it satisfies three conditions, which are defined as
thermal equilibrium, mechanical equilibrium, and phase equilibrium conditions, and they are necessary and sufficient
conditions for the thermodynamic equilibrium of the system (Ansermet and Brechet, 2019).
Therefore, when a system is in thermodynamic equilibrium, it is in thermal equilibrium, mechanical equilibrium, and phase
equilibrium, which indicates that the internal temperature, pressure, and chemical potential are the same everywhere. There is
no heat transfer, water movement (Dexter et al., 2012), and phase transition (Braudeau and Mohtar, 2021). Both the water and
vapor contents are constant at this moment.

#### 83  2.1.2 Water equilibrium and transformation in unsaturated soils

Water in unsaturated soils exists mainly in the form of bound and capillary water. The surface of mineral particles can adsorb
the surrounding water molecules and generate bound water due to van der Waals and electrostatic interactions between mineral
particles and water (Lu and Zhang, 2019; Whalley et al., 2013). Meanwhile, capillary water can be formed locally due to the
interaction force between water molecules (surface tension) (Weil and Brady, 2017a). The bound and capillary water states
can correspond to the equilibrium states of water and vapor inside unsaturated soil.
The system is in non-equilibrium when the water evaporation rate in the local region exceeds the vapor condensation rate. The
relative humidity is less than 100%, and the bound and capillary water is reduced while the vapor increases. The density of
vapor in the region gradually rises as the evaporation process proceeds, resulting in an increase in the probability of vapor
water molecules colliding with each other and the quantity of vapor water molecules absorbed by the water surface, which
indicates the vapor condensation rate accelerating. The system is in thermodynamic equilibrium when the evaporation and
condensation rates are equal, at which time the temperature and the concentration of water and vapor are constant, and the




relative humidity is 100%. In addition, the matrix suction of soil is mechanically balanced with the gravity of the bonded and
capillary water. The system is in non-equilibrium when the condensation rate exceeds the evaporation rate. Vapor is
supersaturated, the relative humidity is greater than 100%, and condensation occurs. The gravity of the bound and capillary
water is greater than the matrix suction of the soil due to the generated condensate, resulting in gravity water of motion.

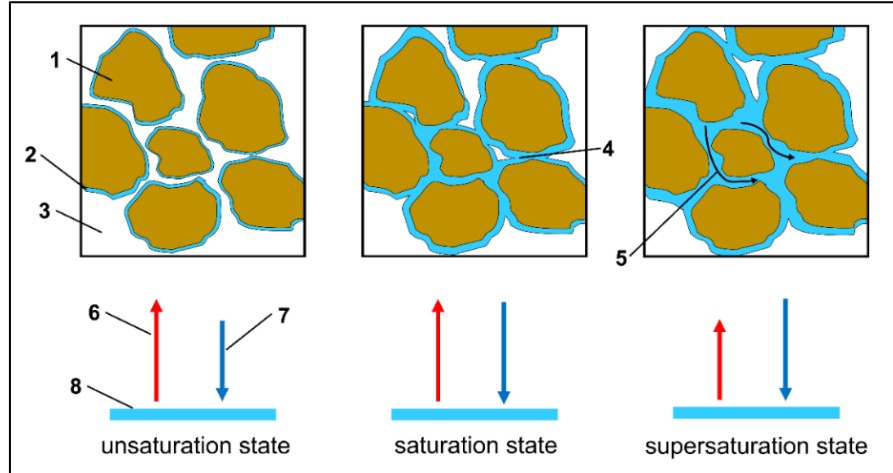


1) Soil particles, 2) Bound water, 3) Pore, 4) Capillary water, 5) Gravity water, 6) Evaporation rate,
7) Condensation rate, 8) Water surface
Fig. 1 The states and transformation of water in relation to equilibrium in unsaturated soils

### 2.2 Water content calculation method

### 2.2.1 Calculating principle

When the unsaturated soil is in thermodynamic equilibrium (100% relative humidity) at a certain temperature, the
concentration of water and vapor is constant, as is the concentration ratio. When the external temperature changes, the
equilibrium state of the system is destroyed, and water and vapor alter their relative content by evaporation or condensation to
generate a new thermodynamic equilibrium state. At this time, the concentration and ratio of water and vapor are constant,
although different from the value of the previous equilibrium state. Therefore, the concentration ratio of water to vapor at
thermodynamic equilibrium is a function of temperature. We can establish the function experimentally, after which the soil
water content can be determined by the function, provided that the soil temperature at thermodynamic equilibrium is obtained.

### 2.2.2 Formula derivation

Water in unsaturated soils exists in liquid and vapor forms at atmospheric temperature, and the masses of the two can be
represented as

$$m_{water}(T) = \rho_{water}(T) \times V_{water}(T) \tag{1}$$



$$m_{vapor}(T) = \rho_{vapor}(T) \times V_{vapor}(T) \tag{2}$$

where $m_{water}(T)$ (kg), $\rho_{water}(T)$ (kg/m³), and $V_{water}(T)$ (m³) represent the mass, density, and volume of water at a temperature of
$T$ (°C), respectively; $m_{vapor}(T)$ (kg), $\rho_{vapor}(T)$ (kg/m³), and $V_{vapor}(T)$ (m³) represent the mass, density, and volume of vapor at the
temperature of $T$ (°C), respectively.
The composition and its concentration are constant when water and vapor in unsaturated soils are in thermodynamic
equilibrium, as shown in Eq. (3).

$$\frac{m_{water}(T)}{m_{vapor}(T)} = \frac{\rho_{water}(T) \times V_{water}(T)}{\rho_{vapor}(T) \times V_{vapor}(T)} = K(T) \tag{3}$$

where $K(T)$ is the thermodynamic equilibrium constant of the system at the temperature of $T$ (°C), and its value is temperature-
dependent.
Assume that $\alpha(T)$ and $\beta(T)$ are the mass and density ratios of water to vapor, respectively, as follows

$$\frac{m_{water}(T)}{m_{vapor}(T)} = \alpha(T) \tag{4}$$

$$\frac{\rho_{water}(T)}{\rho_{vapor}(T)} = \beta(T) \tag{5}$$

The pores of the unsaturated soils are filled with air and water. According to Dalton's law of partial pressure, the volume of
air equals that of vapor. Hence the relationship between the volume of pores, water, and vapor is shown in Eq. (6) (Hillel,
2008b).

$$V_p = V_{water} + V_{vapor} \tag{6}$$

where $V_p$ (m³) denotes the volume of pores in unsaturated soils; $V_{water}$ (m³) denotes the volume of water; and $V_{vapor}$ (m³) denotes
the volume of vapor.
Equation (7) is obtained by combining Eq. (3), Eq. (4), Eq. (5), and Eq. (6),

$$\alpha(T) = \beta(T) \times \frac{V_{water}(T)}{V_p - V_{water}(T)} \tag{7}$$

Therefore, the volumetric water content in the unsaturated soils is

$$W_V(T) = \frac{V_{water}(T)}{V_S} \times 100\% = \frac{\alpha(T) \times V_p}{\alpha(T) + \beta(T)} \times 100\% \tag{8}$$

where $W_v(T)$ (%) represents the volumetric water content in unsaturated soils; $V_s$ (m³) represents the unit volume of soils.
As a result, determining the mass and density ratio of water to vapor is fundamental for calculating soil's volumetric water
content.



## 2.3 The density of water and vapor and their ratio

Research data on the properties of saturated water and vapor at different temperatures have been published by the International
Association for the Properties of Water and Steam (IAPWS) (The International Association for the Properties of Water and
Steam, 2007), Wagner, Kruse (1998), and Kretzschmar et al. (2019), which obtained using equations for the thermophysical
properties of water and steam. The density and density ratio of water and vapor are analysed in this study using the above data.
Although water is the most common liquid in nature, it is also the most unusual, with many peculiar properties, the most
obvious of which is the anomalous density, which is closely related to its complicated molecular structure and configuration
(Gallo et al., 2016; Pettersson et al., 2016). As water molecules are polar, they may be joined by hydrogen bonds to form a
more stable tetrahedral structure, and many water molecules and hydrogen bonds can construct a complex and huge network
(Hillel, 1971; Sciortino and Fornili, 1989).

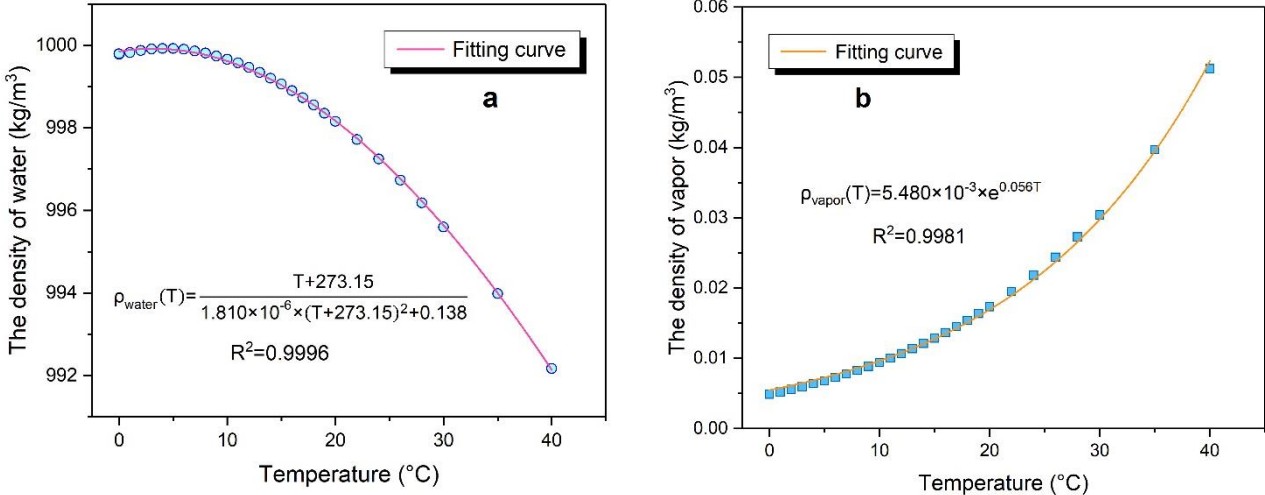

Fig. 2 The density varies with temperature a) water; b) vapor

The structure of water molecules is complicated in the range of 0-4℃, with the coexistence of free molecules, chains, clusters,
hexagonal structures, and ice crystals. In addition, the tetrahedron structure of hydrogen-bonded network of molecules with
larger volume in ice are partially destructed with increasing temperature, which results in water molecules falling into the
cavity of residual tetrahedron (Pang, 2014). Thus, the free volume of water reduces while the density increases and reaches a
maximum at 4℃ (Mahoney and Jorgensen, 2000; Poole et al., 1992). Then, as the temperature continues to rise, the hydrogen
bonding network structure is destroyed, and the distance between water molecules increases, resulting in an increase in water
volume and a decrease in density. According to the study of Yue (1992), the relationship between water density and
temperature is as follows.

$$\rho_{water}(T) = \frac{T + 273.15}{1.810 \times 10^{-6} \times (T + 273.15)^2 + 0.138} \qquad R^2=0.9996 \qquad (9)$$

The greater the temperature, the more intense the thermal movement of water molecules, resulting in more water vapor
molecules to overcome intermolecular forces (such as electrostatic forces, induction forces, dispersion forces, etc.) (Eisenberg





and Kauzmann, 2007), increasing the number of vapor molecules in the air and the density of vapor. Furthermore, vapor
density varies exponentially with temperature, and the fitting equation is as follows.

$$\rho_{vapor}(T) = 5.480 \times 10^{-3} \times e^{0.056T} \qquad R^2=0.9981 \tag{10}$$

Figure 3 shows the variation of density ratio of water to vapor with temperature. Water density fluctuates slightly at
atmospheric temperature, decreasing only from 0.99992 g/cm$^3$ at 4°C to 0.99922 g/cm$^3$ at 40 °C, whereas vapor density varies
exponentially with temperature. Therefore, the density ratio of water to vapor also varies exponentially with temperature, and
the fitting function is

$$\beta(T) = 2.037 \times 10^5 \times e^{-0.064T} \qquad R^2=0.9991 \tag{11}$$

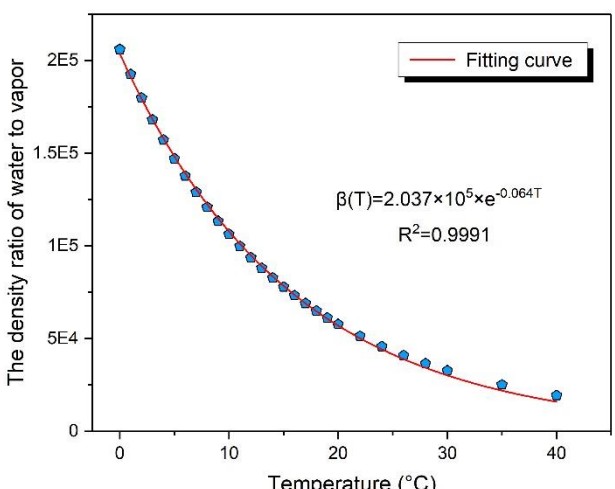


160                  Fig. 3 The density ratio of water to vapor varies with temperature











## 3 Material and methods

### 3.1 Soil samples

The soil samples in this study were collected in Jinan, Shandong Province, China. The soil samples were air-dried, ground, and passed through 30 mesh, 60 mesh, and 200 mesh sieves in turn to produce medium and fine sand, which were then filled into the sandboxes. Soil samples were taken with cutting rings at 5 cm, 15 cm, and 25 cm depths in two sandboxes at the end of the experiment, and the water content, natural density, and dry density were determined by weighing the soil samples before and after drying in an oven at 105 °C for 48 hours (Topp et al., 2007). The gravimetric method with water saturation was used to determine the porosity of soil samples (Flint and Flint, 2002). The properties of medium and fine sand are shown in Table 1.

Table 1 Properties of the two tested soils

| Soil textures | Particle size (mm) | Natural density (g/cm$^3$) | Dry density (g/cm$^3$) | Porosity (%) |
|---|---|---|---|---|
| Medium sand | 0.075 - 0.25 | 1.72 | 1.43 | 35.44 |
| Fine sand | 0.25 – 0.50 | 1.99 | 1.63 | 45.75 |

### 3.2 Experimental setup

To obtain the content of water and vapor at different temperatures at thermodynamic equilibrium in soils, we conducted observing experiments. The experimental apparatus comprises three parts: a water vapor replenishment device, a measurement device, and a data collection device, as shown in Fig. 4. The water vapor replenishment device is a 40 cm × 40 cm × 5 cm top open tank. The measurement device consists of sandboxes and sensors. The dimensions of the sandboxes are 30 cm × 30 cm × 30 cm, and there are 5mm diameter holes distributed in the bottom in the shape of quincunx to allow water vapor from the tank to enter the sandboxes. When filling the sandboxes, the soil is compacted using the water-saturated exhaust method to ensure homogeneity. The temperature and humidity sensors (iButton DS1923, Maxim, USA) measured and recorded the temperature and relative humidity inside the sandboxes. The temperature measurement range is -20-85 °C, with a ± 0.10 °C precision; the relative humidity measurement range is 0-120%, with a ± 0.04% accuracy. The volumetric water content in sandboxes is measured by a soil water content sensor (SYC-SFQ, Saiyasi Technology, China) in the range of 0-100% with ± 2% accuracy. The data collection device (SYD-1, Saiyasi Technology, China) was used to collect and store soil water content. The tanks and sandboxes are made of 5 mm thick polymethyl methacrylate (PMMA).



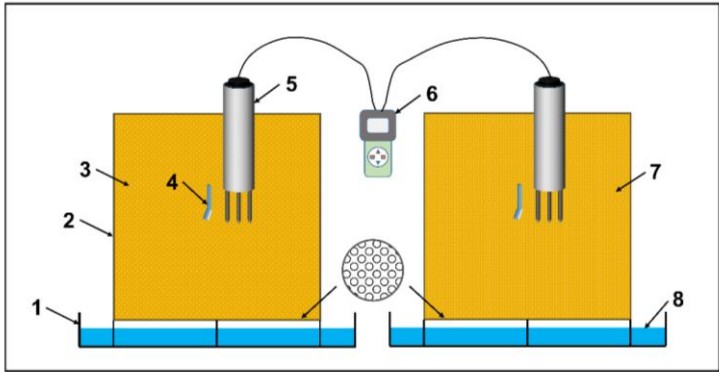

1) Tank, 2) Sandbox, 3) Medium sand, 4) Temperature and humidity sensor, 5) Soil water sensor,

6) Data collector, 7) Fine sand, 8) Water

Fig. 4 Experimental apparatus

### 3.3 Data collection and processing

The experiment was conducted on the roof of the School of Environmental Studies, China University of Geosciences (Wuhan) in a completely open natural environment. Changes in external circumstances produce variations in soil temperature and water content. The experiment began in June 2020 and ended in January 2021. Temperature, humidity, and soil water content were all measured at a rate of one time per minute. More than 100,000 data sets were collected during the experimental period. Nonetheless, the majority of the data sets were collected while the system was in a non-equilibrium state, necessitating data filtering. The data sets at 100% relative humidity were retained for further analysis and calculation.

The collected temperature values were brought into Eq. (9) and Eq. (10) respectively to get the density of water and vapor at that temperature. The volume of water per unit volume of soil is the collected soil water content, and the volume of vapor can be calculated using Eq. (6). The masses of water and vapor at this temperature can be determined according to Eq. (1) and Eq. (2), and their mass ratios can be calculated using Eq. (4).

### 3.4 Significance test

Fisher's Permutation test was performed on the data sets to examine whether there was a significant difference in the mass ratio functions of water to vapor in different media. The principle is to compare whether there is a difference between the coefficients of the fitting functions of the two data sets. The mass ratio functions of water to vapor in medium and fine sand are as follows, respectively.

$$\alpha_1(T) = a_1 e^{b_1 T} \tag{12}$$

$$\alpha_2(T) = a_2 e^{b_2 T} \tag{13}$$





The coefficient differences between the two models are defined as $c = a_1 - a_2$ and $d = b_1 - b_2$, and the null hypothesis of
the test is $H_0$: $c = 0, d = 0$.
The models (12) and (13) were fitted to the data of the medium and fine sand groups, respectively, and the initial estimates of
the coefficients $\hat{a}_1$, $\hat{a}_2$, $\hat{b}_1$ and $\hat{b}_2$, and the coefficients difference ($\hat{c}_0 = \hat{a}_1 - \hat{a}_2$ and $\hat{d}_0 = \hat{b}_1 - \hat{b}_2$) between the two groups
were obtained.
The medium sand data sets ($n_1$ data sets) and the fine sand data sets ($n_2$ data sets) were combined to produce a sample S
consisting of $n_1 + n_2$ observations. From the sample S, $n_1$ observations were randomly selected (without replacement) and
considered as the "medium sand data sets" (denoted as $S_m$), while the remaining $n_2$ observations were considered as the "fine
sand data sets" (denoted as $S_f$). For empirical samples $S_m$ and $S_f$, models (12) and (13) were fitted to obtain $\hat{a}_1^{S_1}$, $\hat{a}_2^{S_1}$, $\hat{b}_1^{S_1}$ and
$\hat{b}_2^{S_1}$ ($S_1$ denotes the first sampling), as well as the coefficients difference ($c^{S_1} = \hat{a}_1^{S_1} - \hat{a}_2^{S_1}$ and $d^{S_1} = \hat{b}_1^{S_1} - \hat{b}_2^{S_1}$) between the
two groups.
Repeating the process of sampling and calculating the coefficient differences 1000 times, the empirical distribution of
coefficients difference $c^{S_j}$, $d^{S_j}$ (j=1, 2, ⋯, 1000) can be determined.
Therefore, the empirical p-value can be expressed as

$$\hat{p}_c = \frac{\#\{c^{S_j} > \hat{c}_0\}}{1000} \tag{14}$$

$$\hat{p}_d = \frac{\#\{d^{S_j} > \hat{d}_0\}}{1000} \tag{15}$$

where $\#\{c^{S_j} > \hat{c}_0\}$ and $\#\{d^{S_j} > \hat{d}_0\}$ denote the number of $c^{S_j}$ and $d^{S_j}$ that are greater than the initial estimates $\hat{c}_0$ and $\hat{d}_0$,
respectively.
The null hypothesis can be rejected at the 5% level if the empirical p-value is less than 0.05, indicating that the coefficients
difference between the two groups of data is significant; otherwise, the coefficients difference is not significantly different at
the 5% level.
All of the above processes were performed in Stata 16.
**4 Results and Discussions**
**4.1 The volume of water and vapor**
Figure 5 and Figure 6 depict the fluctuation in volume and volume proportion of water and vapor within medium and fine sand
at different equilibrium temperatures (The volume of water and vapor is the content of 1m³ of soil). Since the porosity of
medium sand is lower than that of fine sand, the volume of water and vapor in medium sand is less.
The volumes of water and vapor fluctuate with temperature in both medium and fine sand, while the overall change is small,
and the variation is more significant in medium sand than that in fine sand. The volumetric water content of the soil is affected





by temperature, according to Kocarek and Kodesova (2012). We consider that temperature has two effects on volumetric water
content: on the one hand, temperature changes affect the density of water and vapor, resulting in variations in their volume; on
the other hand, it may alter the equilibrium of water and vapor, leading to changes in their relative contents. In this study, the
density of water fluctuates less at atmospheric temperature, and thus its volume changes slightly. Furthermore, due to the
limitations of sensors' precision, the volume changes of water and vapor while they are in equilibrium are challenging to
determine accurately. Therefore, the correlation between temperature and volume of water and vapor is weak.

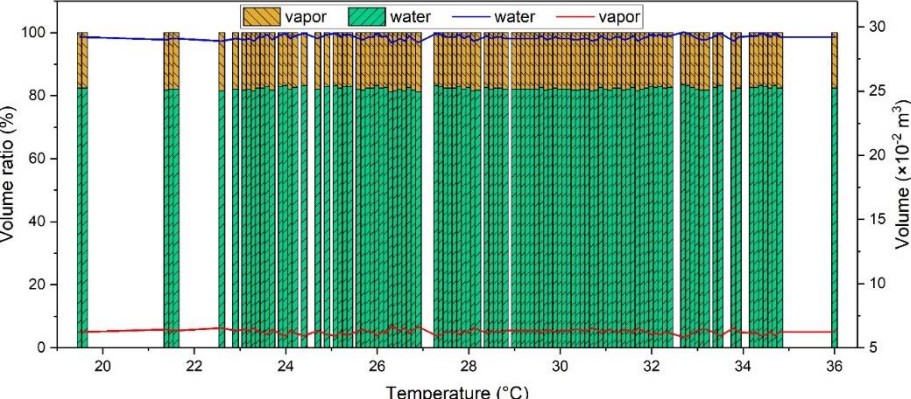


Fig. 5 Variation of the volume of water and vapor with temperature in medium sand

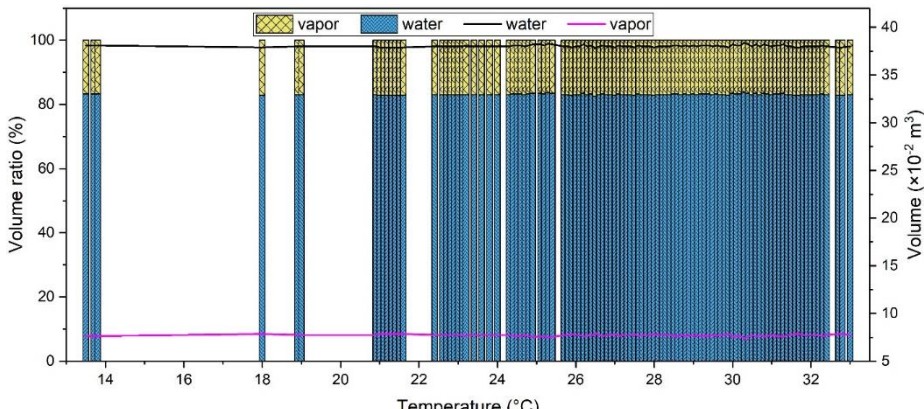


Fig. 6 Variation of the volume of water and vapor with temperature in fine sand
Although there are differences in the volume of water and vapor between medium and fine sand, the volume ratio of the two
is almost consistent, about 83%:17%. As a result, the volume ratio of water to vapor in soils is less dependent on soil properties
at thermodynamic equilibrium.





### 4.2 The mass ratio of water and vapor

### 4.2.1 The mass ratio in different media

The concentration of water and vapor and their mass ratio in unsaturated soil is constant when the system is in thermodynamic
equilibrium. Moreover, temperature variations impact the equilibrium state of water and vapor, leading to changes in their
relative contents. Thus, this section will discuss the mass ratio of water to vapor as a function of temperature. The masses of
water and vapor and their mass ratio are calculated by Eq. (1), Eq. (2), and Eq. (4), respectively. This study obtained 254 sets
of equilibrium state data in medium sand and 231 in fine sand. 90% of the data sets are utilized for the analysis of the mass
ratio of water to vapor in relation to temperature, and the remaining data sets are used for the reliability analysis in section 4.3.
The variation of the mass ratio of water to vapor in medium and fine sand with temperature is depicted in Fig. 7a and Fig. 7b,
respectively. The equilibrium points in medium and fine sand are primarily dispersed in the 22-34 °C temperature range. When
the temperature is low, the intensity of water evaporation is weak, and the water conversion rate into vapor is slow, resulting
in low vapor content in the pore and a non-equilibrium condition. The water conversion rate to vapor increases as the
temperature rises, resulting in an increase in the water vapor content and the probability of an equilibrium condition.

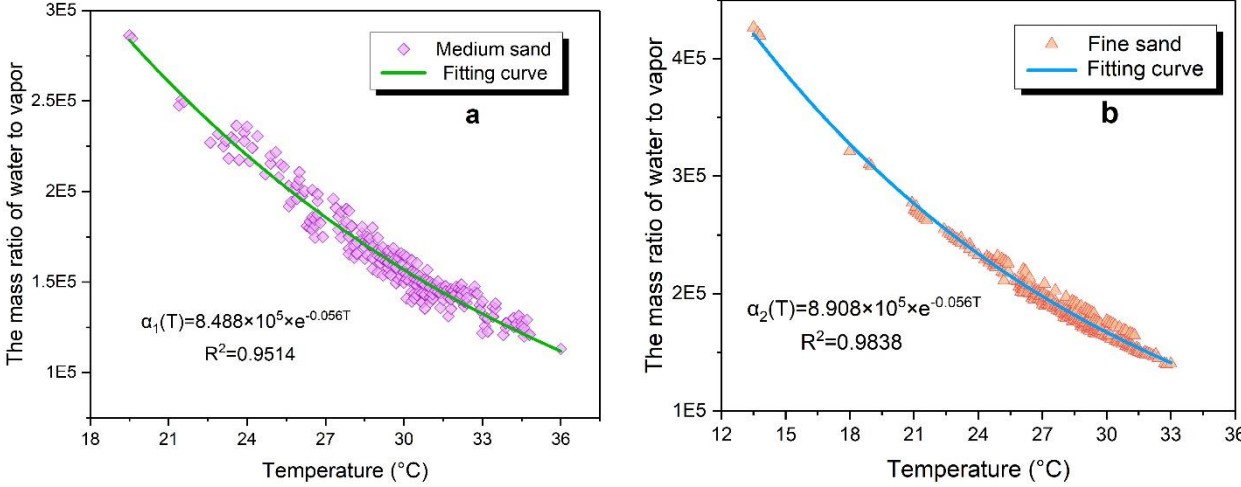

Fig. 7 The mass ratios of water to vapor vary with temperature a) medium sand; b) fine sand

The mass ratios of water to vapor decrease exponentially with increasing temperature. The density of water decreases slightly
as the temperature rises (experiment temperatures are above 4 °C). In contrast, the vapor density increases exponentially, and
the volume ratios of water to vapor in medium and fine sand are almost constant, according to section 4.1. As a result, the
mass ratios of water to vapor vary with temperature similarly to their density ratios, and both decrease exponentially. The
fitting equations for the mass ratio of water to vapor and temperature in medium and fine sand, respectively, are

$$\alpha_1(T) = 8.488 \times 10^5 \times e^{-0.056T} \qquad R^2 = 0.9515 \qquad (16)$$

$$\alpha_2(T) = 8.908 \times 10^5 \times e^{-0.056T} \qquad R^2 = 0.9838 \qquad (17)$$





### 4.2.2 Significance of difference analysis

The difference in specific surface area between the medium and fine sand used in this study is due to a variation in particle size, which resulted in different adsorption forces on water and vapor and moisture content discrepancies (Khlosi et al., 2013). According to the analysis in sections 2.1 and 2.2, when the unsaturated soil is in thermodynamic equilibrium, the system is in mechanical equilibrium, and the mass ratio of water to vapor is constant, at which the adsorption forces are balanced with the gravity of water. Consequently, the mass ratio of water to vapor is the same in medium sand, fine sand, or other media at the same temperature in thermodynamic equilibrium. While the form of the equations for the mass ratios of water to vapor in medium and fine sand determined in this study is consistent, there are differences in the specific coefficients. In this section, we utilize Fisher's Permutation test to examine whether the mass ratio functions of water to vapor in different mediums differ significantly.

Table 2 shows the results of Fisher's Permutation test on the data of the mass ratios of water to vapor in medium and fine sand.

Table 2 Results of the one-way analysis of variance

| Variables | The initial coefficients | | The coefficients difference | Frequency | Empirical $p$-value |
|:---:|:---:|:---:|:---:|:---:|:---:|
| | Medium sand | Fine sand | | | |
| a | $8.488 \times 10^5$ | $8.908 \times 10^5$ | $-4.199 \times 10^4$ | 942 | 0.942 |
| b | -0.0563 | -0.0558 | $-5.577 \times 10^{-4}$ | 728 | 0.728 |

There is no significant difference between the mass ratio and the water-containing media because the empirical p-value is larger than 0.05, indicating that the mass ratio has no connection with the nature of the medium. Therefore, the final mass ratio could be obtained by fitting all of the data on the mass ratio of water to vapor at thermodynamic equilibrium in medium and fine sand (Fig. 8). The fitting equation is as follows.

$$\alpha(T) = 9.286 \times 10^5 \times e^{-0.058T} \qquad R^2 = 0.9598 \qquad (18)$$

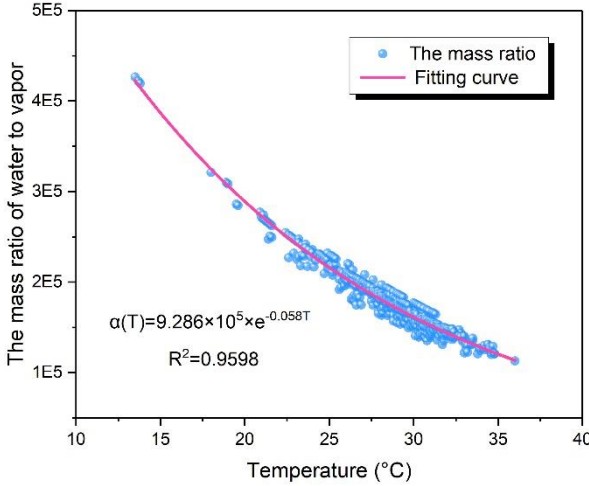

Fig. 8 The mass ratio of water to vapor varies with temperature





## 4.3 Reliability analysis

Sections 2.3 and 4.2 determine the density ratio and mass ratio of water to vapor, respectively, so the water content of unsaturated soil could be calculated using Eq. (8). The remaining equilibrium data sets (i.e., 25 sets of medium sand and 23 sets of fine sand) are utilized for reliability analysis in this section to evaluate the accuracy of the method. The density ratio and mass ratio are calculated using the temperatures in the data sets, and then they are substituted into Eq. (8) to determine the soil volumetric water content, which was then compared to the measured values.

Figure 9 shows the calculated volumetric water content compared to the actual measured. Table 3 shows the statistical results of the absolute errors in the calculated volumetric water content. According to Fig. 9 and Table 3, the absolute errors of the calculated volumetric water content is minimal. The means of the absolute error of the volumetric water content in medium and fine sand are 0.66% and 0.49%, and the maximums are 0.88% and 0.78%, respectively, indicating the method is accurate and reliable. The root mean square error (RMSE) of all data points is 0.61%, and the data points are basically above the 1:1 line, which indicates that the volumetric water content calculated is slightly greater than the measured. Moreover, there are two reasons for the absolute error: the first is the deviation in position of the temperature, humidity, and volumetric water content sensors during the experiment; the second is the error of the experimental apparatus, particularly the volumetric water content sensor, which has a 2% error.

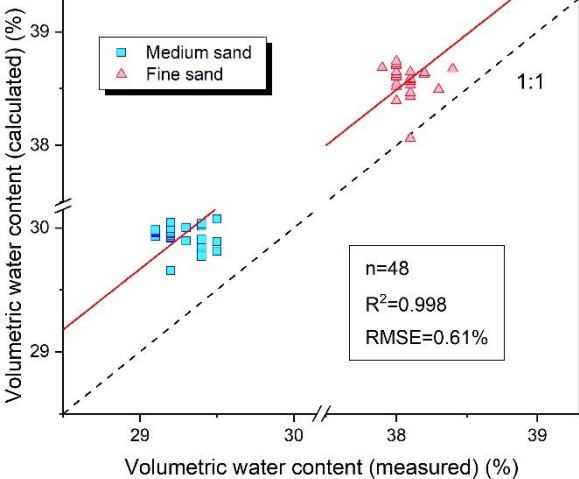

Fig. 9 Comparison of the volumetric water content measured and calculated

Table 3 The statistical results of absolute error of volumetric water content

| Soil textures | Mean/% | Standard deviation/% | Maximum/% | Maximum/% |
|---|---|---|---|---|
| medium sand | 0.66 | 0.18 | 0.88 | 0.31 |
| fine sand | 0.49 | 0.18 | 0.79 | 0.04 |

Furthermore, the standard deviation of absolute error of volumetric water content is 0.18% in both medium and fine sand, indicating a significant degree of dispersion. Then we proceed to analyse the absolute error as a function of temperature. The





absolute error is temperature dependent, as shown in Fig. 10, and tends to increase as the temperature rises. This is because
although the soil volumetric water content fluctuates with temperature, it has no significant correlation. In contrast, the
volumetric water content obtained by this study's method is positively correlated with temperature, resulting in a more
significant absolute error at higher temperatures.

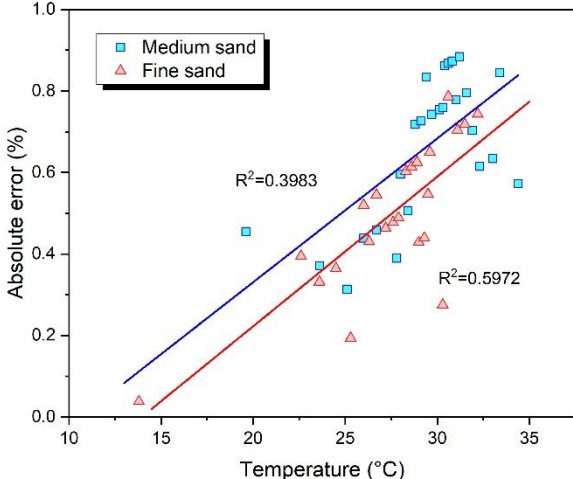

313                        Fig. 10 The absolute error of the volumetric water content varies with temperature

**4.4 Potential application of the method**
The time domain and frequency domain reflection methods are widely used for field soil water content measurement with an
absolute error higher than 2% (Jacobsen and Schjønning, 1993; Lekshmi et al., 2014). The method proposed in this study has
an absolute error of less than 1%, which is advantageous in terms of accuracy.
The method we proposed is applicable to the field as well as the laboratory. The volumetric water content can be determined
with Eq. (8) using the mass ratio by Eq. (18) and the density ratio by Eq. (11), using the measurements of soil porosity (or rock
fissure rate), temperature, and relative humidity. The relative humidity is not used in the estimation of the volumetric water
content but to determine whether the geotechnical system is in thermodynamic equilibrium. Only systems that are in
equilibrium or near equilibrium can use this method to estimate the water content.
To verify the applicability of the method, we conducted field experiments. We selected a woodland in Wuhan and placed
temperature and humidity monitors at different depths of the soil (0 cm, 10 cm, 20 cm, 40 cm, 80 cm, 100 cm), collecting
temperature and humidity data every 5 minutes. We found that the relative humidity in the soil under natural conditions in the
field was mostly concentrated at 99%-101% after one week of continuous measurement, which indicates that the water and
vapor in the soil are in near equilibrium under natural conditions, so the soil water content can be determined directly using
the method we proposed. The results are shown in Table 4.






Table 4 The measurement results of soil at different depths

|  | 0 cm | 10 cm | 20 cm | 40 cm | 80 cm | 100 cm |
|---|---|---|---|---|---|---|
| Temperature/°C | 19.37 | 19.45 | 19.98 | 20.47 | 21.65 | 21.79 |
| Relative humidity/% | 100.9 | 99.6 | 97.0 | 98.5 | 100.4 | 100.4 |
| Water content/% | 27.78 | 27.78 | 27.79 | 27.81 | 27.84 | 27.84 |

The great potential application for this method is determining the water content of rocks. Water content measurements in soil
are fairly mature, whereas it is still more complicated in rocks due to sampling difficulties and near-closed fissures. The above
studies have confirmed that the density and mass ratios of water to vapor are independent of the nature of the water-containing
medium, therefore the method is applicable to any medium. We drilled a hole that has a diameter of 10 cm, a depth of 200 cm
in the rock wall of an abandoned quarry in Jinan, Shandong Province. Temperature and humidity monitors were also placed
at different depths (0 cm, 10 cm, 20 cm, 40 cm, 60 cm, 100 cm, and 150 cm) to collect temperature and humidity data in the
rock. The results showed that the relative humidity in the rock was also concentrated at 99%-101%, so the method can be
directly applied to calculate the water content in the rocks. The water content within each depth of the rock was calculated to
be approximately 0.612%, and the result would be validated subsequently.
Table 5 The measurement results of rock at different depths

|  | 0 cm | 10 cm | 20 cm | 40 cm | 60 cm | 100 cm | 150 cm |
|---|---|---|---|---|---|---|---|
| Temperature/°C | 16.70 | 17.93 | 17.85 | 17.90 | 18.00 | 18.65 | 19.53 |
| Relative humidity/% | 43.8 | 99.3 | 100.6 | 100.9 | 100.6 | 100.2 | 100.1 |
| Water content/% | / | 0.612 | 0.612 | 0.612 | 0.612 | 0.612 | 0.613 |

According to our monitoring and investigation in North and Central China, soils and rocks up to a depth of 100 cm and 150
cm, respectively, are in equilibrium or near equilibrium under natural conditions, so the volumetric water content of soils and
rocks can be estimated using this method. Nevertheless, the application of the method will be restricted in some areas with dry
climates (such as northwest China and the western United States), where the water content in rock-soil mass and air is low and
the system is non-equilibrium the vast majority of the time. In this situation, it is necessary to choose the appropriate time for
measuring the water content of the rock-soil mass. A temporary equilibration of the system occurs between dusk and dawn,
when the ambient temperature decreases, resulting in a condensation process in rock-soil mass. It is the proper time to
determine water content.
**5 Conclusions**
In this study, we proposed an alternative method to measure soil water content using the ratios of mass and density of water to
vapor based on system science and thermodynamics. We derived the mass ratio and density ratio of water to vapor as a function
of temperature from published data and validated the accuracy of the proposed method using observed temperature, relative





humidity, and volumetric water content in two soil textures of medium and fine sand. We further demonstrated that the mass
ratio function is independent of the nature of the water-containing media using theoretical and statistical analysis. The absolute
error of soil water content between the calculated and measured ones is less than 1% and is positively correlated with
temperature. When the geotechnical system is in equilibrium or near equilibrium, the water content can be calculated by the
method we proposed. According to monitoring and investigation, soils and rocks up to a depth of 100 cm and 150 cm,
respectively, are in equilibrium or near equilibrium under natural conditions, so the volumetric water content of rock-soil mass
can be estimated using this method. This study significantly improves the measurement accuracy of soil water content,
eliminates the effect of water-containing medium on soil water content measuring, and has excellent potential for application
in the field soil and even rock water content measurement.
**Data availability**
The data that support the findings of this study are available from the corresponding author upon reasonable request.
**Author contribution**
**Danhui Su**: Conceptualization, Data curation, Writing - original draft. **Jianwei Zhou**: Methodology, Conceptualization,
Writing – review & editing. **Zuocong Yin**: Methodology, Investigation, Formal analysis. **Haibo Feng**: Methodology,
Investigation, Writing – review & editing. **Xiaoming Zheng**: Investigation, Validation. **Xu Han**: Investigation. **Qingqiu Hou**:
Investigation.
**Competing interests**
All co-authors have seen and agree with the contents of the manuscript. We declare that we have no known competing financial
interests or personal relationships that could have appeared to influence the work reported in this paper.
**Acknowledgments**
The authors gratefully acknowledge financial support from the National Natural Science Foundation of China (No. 42077182).
We are also extremely grateful to Prof. Hengli Xu for assistance on the experimental design.

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
