# Peer review of "A calculation method of unsaturated soil water content based on 2 thermodynamic equilibrium"

_Hydrology and Earth System Sciences, 2023_

## Referee Comment (RC1)

The manuscript by Danhui Su et al. proposed a method to determine unsaturated soil water content based on thermodynamic equilibrium, which is independent of the water containing media and with the absolute error less than 1%. The manuscript presented the relationship between soil water content and the mass ratio and density ratio of water to vapour at thermodynamic equilibrium, conducted sandbox experiments to validate the function of the mass ratio and density ration of water to vapour.

This study is of potential application in the monitoring soil water content, even for the rocks. However, there are some concerns from my perspective. First, regarding the validation of the proposed method with the measured soil volumetric water content, the measured "truth" is obtained using a soil water content sensor, with an accuracy of 2%. I am not sure how much you believe in the measurements of the selected soil water content sensor. Have you calibrated the used soil water content sensor in terms of different soil textures (medium sand and fine sand in this manuscript)? Second, there are a wide range of soil textures, why only selected the medium sand and fine sand, how about the clay soils and organic soils?

Overall, the idea of this manuscript is interesting, but the validation part is not that convincing from my perspective. My specific comments are as follows.

Abstract:

Line 10-11: you mentioned the current challenges of the measurement methods of soil water content, this manuscript dealt with all of them or not. Please clarify.

Introduction:

Line 66-68: "Furthermore, the mass ratio function is independent of the nature of the water-containing media, according to theoretical analysis and statistical results. The absolute error of soil water content between the calculated and measured is less than 1% and is positively correlated with temperature."
This is the findings of your work, should not belong here.

Results:

"When the temperature is low, the intensity of water evaporation is weak, and the water conversion rate into vapor is slow, resulting in low vapor content in the pore and a non-equilibrium condition."
here you mean, the temperature is low corresponds to a non-equilibrium condition?? Please clarify.

Line 302: "volumetric water content sensor, which has a 2% error."
here the measured volumetric water content is obtained using the sensors?
the absolute error, is assumed that the used sensor is the truth, which is not according to your words (2% error).
Please give more evidence demonstrating the validity of the selected soil water sensor, e.g., calibration work.

Line 308-311: "This is because although the soil volumetric water content fluctuates with temperature, it has no significant correlation. In contrast, the volumetric water content obtained by this study's method is positively correlated with temperature, resulting in a more significant absolute error at higher temperatures."
here you mean that it is the nature that soil water content should be positively correlated with temperature? Please clarify.

Line 357: "When the geotechnical system is in equilibrium or near equilibrium, …"
how to determine whether the geotechnical system is in equilibrium or near equilibrium or not in practical?

Figure 4. Please give the dimensions in details and the position of soil sensors.
Figure 5:
legend: what is the difference between two "water"?
Figure 5 and figure 6 can be merged into one figure.

Figure 9:
From this figure, the determination coefficient is 0.998, largely due to the two sets of data with different values. and the fitting trend line indicate that the calculated water content is overall overestimated. Thus, although the determination coefficient is high, the validity of the calculated method is not convincing. Please explain how do you deal with the overestimation and dispersion of results?

Table 4 and Table 5:
In these two application cases, the values of water content are relatively stable, i.e., around 27.8% and 0.612% for the soil and rock, respectively. Please explain why the water content is kind of stable at different depths.

I suggest the authors add the limitations of the proposed method, e.g., whether it can be applied in frozen soils, not suitable for measuring soil water content at the surface soil layers (usually non equilibrium), etc.

Although I am not the native English speaker, I think the manuscript can be improved from the perspective of English

---

## Author Comment (AC1)

**Response to Reviewers' Comments for HESS Discussion Article**

**Title:** A calculation method of unsaturated soil water content based on thermodynamic equilibrium

**Author(s):** Danhui Su et al.

**MS No.:** hess-2023-44

**MS type:** Research article

Dear Referee #1,

We greatly appreciate you providing valuable and constructive comments on our manuscript hess-2023-44. We considered each comment and will revise/improve the manuscript accordingly. The individual comments are replied to below. In the following the reviewer comments are formatted in black font and our responses are in blue.

This study is of potential application in the monitoring soil water content, even for the rocks. However, there are some concerns from my perspective. First, regarding the validation of the proposed method with the measured soil volumetric water content, the measured "truth" is obtained using a soil water content sensor, with an accuracy of 2%. I am not sure how much you believe in the measurements of the selected soil water content sensor. Have you calibrated the used soil water content sensor in terms of different soil textures (medium sand and fine sand in this manuscript)? Second, there are a wide range of soil textures, why only selected the medium sand and fine sand, how about the clay soils and organic soils?

Overall, the idea of this manuscript is interesting, but the validation part is not that convincing from my perspective. My specific comments are as follows.

We thank you for providing critical comments and pointing out your concerns about our validation method and soil texture selection.

In response to your first question, we verified the accuracy of the soil water content sensors before the experiment started, and the detailed process and results are shown in the response to question 4.

In response to your second question, we conducted a pre-experiment before the official experiment began. We selected four soil textures, coarse sand (0.50-2.00 mm), medium sand (0.25-0.50 mm), fine sand (0.075-0.25 mm), and silty sand (<0.075 mm), which were filled in sandboxes and placed temperature and humidity sensors to record the temperature and humidity data and observe the conversion of water and vapor in the sandboxes. The results showed that although the coarse sand had the smallest porosity, it had more large pores and exchanged moisture closely with the air, so the capillary water in the soil was quickly evaporated away and the water in the bottom tank could not enter the soil quickly, resulting in data collected at thermodynamic equilibrium that were not statistically significant.

In contrast, although the silty sand was more porous, it had a higher clay particle content and a smaller effective porosity. Therefore, the internal processes of water and vapor migration and transformation were slow, and the water content changed less in a short period, resulting in data collected at thermodynamic equilibrium that was not statistically significant.

Clay soils have a higher clay particle content than silty sand, and their permeability is worse. It is often regarded as a water barrier in hydrogeology. Therefore, although the experiments were not carried out using clay soils in this study, the results are predictably worse compared to silty sand. Consequently, medium and fine sands with suitable water-holding capacity and permeability were used as the subjects of this study.

It is well known that soil organic matter can contribute to the formation of agglomerates and colloids and enhance the ability of soil to adsorb water, thus affecting soil water content (Bircher et al., 2012; Roth et al., 1992; Topp et al., 1980). Soil organic matter may affect the conversion of water and water vapor within the soil, but it cannot affect the equilibrium state of the two. This is

because, for a thermodynamic equilibrium system, the mass ratio of water to vapor is determined at a certain defined temperature. Therefore, organic soils were not considered in this study. The method can be examined and validated in organic soils afterward.

1. Line 10-11: you mentioned the current challenges of the measurement methods of soil water content, this manuscript dealt with all of them or not. Please clarify.

Thank you for your comments. The most important problem solved in this study is the elimination of the effect of the water-containing medium on the water content test. Theoretically, this method can be used to determine the water content in sand, clay, organic soils, and even rocks.

Currently, the operation of our measurement method is not simple. It is necessary to obtain the temperature and humidity values in the water-containing medium, after which the porosity of the soil or the fracture rate of the rock is measured and finally brought into the formula to calculate the water content.

From the perspective of experimental results, the error of water content between the calculated and measured ones is less than 1%, which is less than the measurement accuracy of most methods.

The development of the physical product is in progress simultaneously. It is expected that the purpose of simplifying measurement steps and reducing the cost of detection can be achieved in the future.

2. Line 66-68: "Furthermore, the mass ratio function is independent of the nature of the water-containing media, according to theoretical analysis and statistical results. The absolute error of soil water content between the calculated and measured is less than 1% and is positively correlated with temperature."

This is the findings of your work, should not belong here.

Thank you very much for your valuable suggestions. We will adjust this section in the revised manuscript.

3. "When the temperature is low, the intensity of water evaporation is weak, and the water conversion rate into vapor is slow, resulting in low vapor content in the pore and a non-equilibrium condition."
here you mean, the temperature is low corresponds to a non-equilibrium condition? Please clarify.

Thank you for your comments. The main condition for reaching thermodynamic equilibrium in an open soil system is the temperature change.
We counted the variation of temperature and saturation frequency during the day (Figure. R1). The temperature showed fluctuating changes. The temperature was at its lowest value at 8:00, after which it gradually increased and reached its maximum value at 16:00; subsequently, it decreased again until 8:00 the next day. This corresponded to the changes in saturation frequency. The saturation frequency was slightly lower when the temperature reached its maximum at 16:00, after which it increased as the temperature decreased. This process was because the saturation absolute humidity decreased during the temperature decrease, while the absolute humidity did not change, resulting in an increase in relative humidity and, therefore, an increase in saturation frequency. However, the decrease in temperature would not make the saturation frequency increase continuously. When the temperature decreased to the dew point temperature, the saturated state in the sandboxes would change to the supersaturated state, i.e., condensate was generated. The vapor in the sandboxes was consumed and the saturation frequency would decrease, i.e., the state of 11:00 to 8:00. Then, the temperature gradually increased and

the condensate evaporated to generate vapor, which made the vapor content replenished and the saturation frequency gradually increased. With the enhancement of solar radiation, the vapor in the sandboxes gradually evaporated, while the vapor content decreased due to the low recharge of the tank to the soil, resulting in a slightly lower saturation frequency, i.e., the state from 13:00 to 16:00.

This result was consistent with that in the manuscript. When the temperature decreased, the frequency of equilibrium increased, but a continuous decrease in temperature, on the contrary, led to a decrease in the frequency of equilibrium due to supersaturation consumption of vapor. Therefore, the system might be more non-equilibrium at low temperatures.

[Figure]

Figure. R1 Variation of temperature and saturation frequency with time

4. Line 302: "volumetric water content sensor, which has a 2% error."

here the measured volumetric water content is obtained using the sensors?

the absolute error, is assumed that the used sensor is the truth, which is not according to your words (2% error).

Please give more evidence demonstrating the validity of the selected soil water sensor, e.g., calibration work.

Thank you for your comments. The measured volumetric water content in section 4.3 was obtained using the sensors. The soil water content sensors we used are pre-calibrated by the sensor manufacturer and have four built-in calibration curves for standard (normal) soils (density of 1.4 g/cm$^3$), low density soils (density less than or equal to 1.1 g/cm$^3$), high density soils (density greater than or equal to 1.7 g/cm$^3$), and peaty soils (density of 0.45 g/cm$^3$). After pre-calibration, the sensors are accurate to ± 0.3%.

However, the calibration and testing of the sensors were performed under laboratory conditions and were not specific to particular soil textures. Therefore, we verified the accuracy of the sensor before starting the experiment. We took three soil samples in each of the two sandboxes and measured the water content of each sample with sensors. After that, the samples were put into the oven and dried at 105°C for 48 hours and then weighed respectively. The results are shown in Table R1.

Table R1 Comparison of sensors and oven drying for measuring soil moisture content

| Testing method | Medium sand | | | Fine sand | | |
|---|---|---|---|---|---|---|
| Oven-drying | 29.2% | 29.1% | 29.1% | 37.1% | 39.8% | 35.9% |
| Sensors | 28.6% | 29.6% | 29.5% | 38.2% | 37.9% | 38.4% |

The results indicated that in the medium sand with low moisture content, the error of the sensor was small, within 1%; in the fine sand with high moisture content, its error increased, with an average of 1.8%. It basically meets the experimental requirements.

In addition, we have recognized the importance of instrument calibration, and we will add a supplemental instrument calibration experiment for medium and fine sand, respectively, to calculate and verify the water content with the

calibrated data. We will add that part in the revised manuscript.

5. Line 308-311: "This is because although the soil volumetric water content fluctuates with temperature, it has no significant correlation. In contrast, the volumetric water content obtained by this study's method is positively correlated with temperature, resulting in a more significant absolute error at higher temperatures."
here you mean that it is the nature that soil water content should be positively correlated with temperature? Please clarify.

Thank you for your comments. Ni et al. (2019) showed that an increase in temperature causes an increase in soil evapotranspiration and thus a decrease in soil water content. It has also been shown that temperature affects the performance of the sensor, resulting in an increase in the measured soil water content with temperature (Schwartz et al., 2019).
The results of our experiments of indoor and field monitoring showed that there is no significant correlation between temperature and soil water content. The equation we proposed to calculate the water content is simplified as shown in Eq. R1.

$$W_V(T) = \frac{V_p}{1 + 0.219e^{-0.006T}} \qquad (R1)$$

From Eq. R1, it can be concluded that the calculated water content increases slightly with increasing temperature. Therefore, the implication in the manuscript is that the error of this method might increase at high temperatures.

6. Line 357: "When the geotechnical system is in equilibrium or near equilibrium, …"
how to determine whether the geotechnical system is in equilibrium or near equilibrium or not in practical?

Thank you for your comments. This is an important issue because, in the natural environment, thermodynamic equilibrium is relative, specific, temporary, and local, while non-equilibrium is absolute, universal, eternal, and global.

As Line 105 shows, when the geotechnical system is in equilibrium, the relative humidity of the internal system is 100%. Therefore, in practice, a humidity sensor is needed to determine whether the geotechnical system is in thermodynamic equilibrium. Although natural systems are often in non-equilibrium, our field monitoring data showed that rocks and soils in central and northern China were in near-equilibrium (relative humidity concentrated in the range of 99%-101%) at depths of 10-150 cm, and the water content calculated under these conditions was almost identical to that in equilibrium. Therefore, the water content can also be calculated by this method in near-equilibrium conditions.

7. Figure 4. Please give the dimensions in details and the position of soil sensors.

Figure 5:

legend: what is the difference between two "water"?

Figure 5 and figure 6 can be merged into one figure.

Thank you very much for your valuable suggestions. We have made the following changes to Figure. 4 and Figure. 5:

Figure. 4: The detailed dimensions of the experimental apparatus and the position of the soil sensors have been marked in Figure. R2.

Figure 5: I am very sorry that I made you doubt due to my unclear marking in the figures. In Figure 5 and Figure 6, the column legend indicates the volume ratio of water and vapor, and the line legend indicates the volume of water and vapor. We have made modifications in Figure. R3 and Figure. R4.

We have tried to merge Figure 5 with Figure 6 before. However, there were

many data in the two sandboxes with close temperatures, and if they are merged, the columns would overwrite each other and would cause data loss. Therefore, we finally showed the two figures separately and hope you understand our decision.

[Figure]

Figure. R2 Experimental apparatus

[Figure]

Figure. R3 Variation of the volume of water and vapor with temperature in medium sand

[Figure]

Figure. R4 Variation of the volume of water and vapor with temperature in fine sand

8. Figure 9:

From this figure, the determination coefficient is 0.998, largely due to the two sets of data with different values. and the fitting trend line indicate that the calculated water content is overall overestimated. Thus, although the determination coefficient is high, the validity of the calculated method is not convincing. Please explain how do you deal with the overestimation and dispersion of results?

Thank you for your comments. Although the calculated water content is higher than the measured value, the absolute error is less than 1% (0.66% for medium sand and 0.49% for fine sand), which is more accurate than most measurement methods.

The reason for the higher determination coefficient is that the absolute errors of the two sets of data are relatively close to each other, indicating that the accuracy is higher in both different water-containing mediums.

The important reason for dispersion is temperature, as shown in Figure. 10 and in Lines 306-311. The water content calculated by this method increases slightly with increasing temperature and therefore the error is large at high temperatures.

We believe that overestimation and dispersion can be an important foundation for calibrating the method.

9. Table 4 and Table 5:

In these two application cases, the values of water content are relatively stable, i.e., around 27.8% and 0.612% for the soil and rock, respectively. Please explain why the water content is kind of stable at different depths.

Thank you very much for pointing out the problem in our manuscript. Due to negligence on our part, when calculating the soil water content in Table. 4, the porosity of 0-10 cm was incorrectly brought into the equation to calculate the water content at all depths, resulting in almost the same water content at different depths. As shown in Table. R2 and Figure. R5, we have corrected the error and will update this section in the revised manuscript.

The difficulty of collecting in situ samples in rocks results in fracture rates not being readily available. Therefore, we assumed that fractures were homogeneously developed within the rocks and calculated the water content at different depths by measuring the fracture rate in the surface layer of the rocks and bringing it into Eq. (8). This resulted in relatively close water content of the rocks at different depths. This section is intended to show the potential application of this method in measuring the water content of rocks. For later applications, 3D laser scanning, downhole video or other methods can be used to obtain fracture rates at different depths, thus making the results more accurate.

Table. R2 The measurement results of soil at different depths

|                     | 0 cm  | 10 cm | 20 cm | 40 cm | 80 cm | 100 cm |
|---------------------|-------|-------|-------|-------|-------|--------|
| Temperature/°C      | 19.37 | 19.45 | 19.98 | 20.47 | 21.65 | 21.79  |
| Relative humidity/% | 100.9 | 99.6  | 97.0  | 98.5  | 100.4 | 100.4  |
| Water content/%     | 27.78 | 27.78 | 26.62 | 25.71 | 24.99 | 24.74  |

[Figure]

Figure. R5 Variation of soil water content at different depths with time

10. I suggest the authors add the limitations of the proposed method, e.g., whether it can be applied in frozen soils, not suitable for measuring soil water content at the surface soil layers (usually non equilibrium), etc.

Thank you very much for your valuable suggestions. The temperature range of this study is 0-40°C, and we only carried out the study when the two phases of water and vapor are in equilibrium. When the temperature is below 0 °C, the coexistence of solid (ice), liquid (water), and gas (vapor) will occur, and its equilibrium mechanism and equilibrium conditions are obviously different from two-phase equilibrium, therefore, this method is not applicable to the measurement of water content in frozen soils.

Furthermore, the application of the method will be restricted in some areas with dry climates and the surface layer of the rock and soil (0-10 cm), where the water content is low and the system is non-equilibrium the vast majority of the time. In this situation, it is necessary to choose the appropriate time for measuring the water content of the rock-soil mass. A temporary equilibration of the system occurs at dusk and dawn, when the ambient temperature decreases, resulting in a condensation process in rock-soil mass. It is the proper time to

determine water content.

We will add this section in the revised manuscript.

11. Although I am not the native English speaker, I think the manuscript can be improved from the perspective of English.

Thank you very much for your valuable suggestions. We will seek native English speakers to help us revise the language of the manuscript to improve the quality of the paper.

Reference

Bircher, S., Skou, N., Jensen, K.H., Walker, J.P., Rasmussen, L., 2012. A soil moisture and temperature network for SMOS validation in Western Denmark. Hydrol. Earth Syst. Sci., 16(5): 1445-1463. DOI:10.5194/hess-16-1445-2012

Ni, J., Cheng, Y., Wang, Q., Ng, C.W.W., Garg, A., 2019. Effects of vegetation on soil temperature and water content: Field monitoring and numerical modelling. J Hydrol, 571: 494-502. DOI:https://doi.org/10.1016/j.jhydrol.2019.02.009

Roth, C.H., Malicki, M.A., Plagge, R., 1992. Empirical evaluation of the relationship between soil dielectric constant and volumetric water content as the basis for calibrating soil moisture measurements by TDR. Journal of Soil Science, 43(1): 1-13. DOI:https://doi.org/10.1111/j.1365-2389.1992.tb00115.x

Schwartz, M., Li, Z., Sakaki, T., Moradi, A., Smits, K., 2019. Accounting for Temperature Effects on the Performance of Soil Moisture Sensors in Sandy Soils. Soil Science Society of America Journal, 83(5): 1319-1323. DOI:https://doi.org/10.2136/sssaj2019.05.0161

Topp, G.C., Davis, J.L., Annan, A.P., 1980. Electromagnetic determination of soil water content: Measurements in coaxial transmission lines. Water Resour Res, 16(3): 574-582. DOI:https://doi.org/10.1029/WR016i003p00574

---

## Author Comment (AC2)

**Response to Reviewers' Comments for HESS Discussion Article**

**Title:** A calculation method of unsaturated soil water content based on thermodynamic equilibrium

**Author(s):** Danhui Su et al.

**MS No.:** hess-2023-44

**MS type:** Research article

Dear Referee #2,

We greatly appreciate you providing valuable and constructive comments on our manuscript hess-2023-44. We considered each comment and will revise/improve the manuscript accordingly. The individual comments are replied to below. In the following the reviewer comments are formatted in black font and our responses are in blue.

Based on the assumption of thermodynamic equilibrium, the authors propose an innovative approach to calculate volumetric water content. In general, the manuscript is well written, and could be accepted for publication if the following points can be addressed sufficiently:

Thank you for your encouraging and constructive comments. Your comments are replied to below.

1. The author discussed various issues of using different measurement techniques for measuring soil water content in the introduction. However it is not clear to this reviewer what is the measuring principle of the soil moisture sensors they used. Could the author help to clarify why they choose particularly this sensor?

Thank you for your comments. The soil moisture sensors we used in this study

are based on the principle of reflection in the frequency domain to measure soil water content. The measurement is made by launching electromagnetic waves and using the relationship between the velocity of the electromagnetic waves in the soil and the soil water content. We chose this sensor because of its fast and stable testing capability, and acceptable measurement error.

2. One major concern this reviewer has is the wide application of this method to monitor the dynamic change of soil moisture content in the field. Please the author help to add a time series plot to clarify this.

Thank you for your comments. We have added three figures (Figure R1, Figure R2, Figure R3) to illustrate the application of this method for monitoring soil moisture dynamic change in the field. We conducted field monitoring experiments in Wuhan, China. Temperature and humidity sensors were placed at different depths in the soil and samples were taken to test the porosity at the corresponding monitoring depth. The monitoring experiment was then started and continued for one week.

Figure. R1 showed the variation of temperature with time at different depths of the soil. The temperature at all depths had an overall steady decrease during the monitoring period. Temperatures in the shallow part (10-40 cm) had daily fluctuations, and the volatility decreased gradually with the increase in depth. Temperatures at depth (80-100 cm) remained stable with almost no fluctuations.

Figure. R2 showed the variation of relative humidity with time at different depths of the soil. The relative humidity at each depth increased rapidly at the beginning of the experiment, after which it fluctuated from 95% to 100% in the shallow part (10-40 cm) and remained stable at about 100% in the deep part (80-100 cm).

Figure. R3 showed the variation of soil water content at different depths with time. Compared to temperature and relative humidity, soil water content fluctuated insignificantly, having a slight daily variation in the shallow part (0-40

cm) and remaining stable in the deep part (80-100 cm).

The results indicate that the method can be well applied to monitor the dynamic changes of soil water content in the field.

[Figure]

Figure. R1 Variation of temperature with time at different depths of the soil

[Figure]

Figure. R2 Variation of relative humidity with time at different depths of the soil

[Figure]

Figure. R3 Variation of soil water content at different depths with time

3. Although the thermodynamic equilibrium conditions exist in the field, it can be frequently interrupted by rainfall events. This reviewer is curious about how this approach could be extended to non-equilibrium conditions, which is more the case in natural environments and the soil moisture of which is more important to be monitored.

Thank you for your comments. This is an important issue because, in the natural environment, thermodynamic equilibrium is relative, specific, temporary, and local, while non-equilibrium is absolute, universal, eternal, and global.

Although natural systems are often in non-equilibrium, our field monitoring data showed that rocks and soils in central and northern China were in near-equilibrium (relative humidity concentrated in the range of 99%-101%) at depths of 10-150 cm, and the water content calculated under these conditions was almost identical to that in equilibrium. Therefore, the water content can also be calculated by this method in near-equilibrium conditions.

4. Although there are supersaturation condition wherein relative humidity will be larger than 100%, I am wondering how this could be the case in the natural environment? Please the authors revisit line 97

Thank you for your comments. There are some important factors influencing the generation of condensation phenomena: temperature, water vapor content, condensation nucleus, etc. At a certain temperature, when the water vapor content exceeds the saturated water vapor content, condensation is theoretically generated, thereby the relative humidity reducing or maintaining at 100%. When there is a lack of condensation nucleus in the air, however, condensation does not occur immediately, and at that time, the relative humidity gradually increases and exceeds 100%. This phenomenon occurs in natural environments such as the atmosphere (Dessler and Sherwood, 2000), soils (Assouline and Kamai, 2019), and rocks (Ho, 1997).

minor technical points
The section 2.2.2 does not consider the impact of osmotic potential, Please the authors help clarify.

Thank you very much for your valuable suggestions. As we all know, water potential is an important parameter that determines the movement of water in unsaturated soils, and it consists of gravitational potential, matrix potential, and osmotic potential. We are very sorry that due to our negligence, the role of the osmotic potential had been omitted from the manuscript. We will add this section in the revised manuscript.

Some minor technical points:

What do you mean by natural density, is it bulk density?

I am very sorry that I was not clear in my words and made you doubt. The natural density in the manuscript means the density of the soil after filling and draining, including soil particles and moisture. I will modify this part in the revised manuscript.

Figure 6, you have two legends both called vapor (and water), please make clear one (column) is for volume ratio, and the other (line) is for volume.

I am very sorry that I made you doubt due to my unclear marking in the figures. In Figure 5 and Figure 6, the column legend indicates the volume ratio of water and vapor, and the line legend indicates the volume of water and vapor. We have made modifications in Figure. R4 and Figure. R5.

[Figure]

Figure. R4 Variation of the volume of water and vapor with temperature in medium sand

[Figure]

Figure. R5 Variation of the volume of water and vapor with temperature in fine sand

Reference

Assouline, S., Kamai, T., 2019. Liquid and Vapor Water in Vadose Zone Profiles Above Deep Aquifers in Hyper-Arid Environments. Water Resour Res, 55(5): 3619-3631. DOI:https://doi.org/10.1029/2018WR024435

Dessler, A.E., Sherwood, S.C., 2000. Simulations of tropical upper tropospheric humidity. Journal of Geophysical Research: Atmospheres, 105(D15): 20155-20163. DOI:https://doi.org/10.1029/2000JD900231

Ho, C.K., 1997. Evaporation of pendant water droplets in fractures. Water Resour Res, 33(12): 2665-2671. DOI:https://doi.org/10.1029/97WR02489